


# Phytoplankton responses to iron and macronutrient fluxes from subsurface waters in the western North Pacific in summer

Huailin Deng[1,2], Koji Suzuki[2,3], Ichiro Yasuda[4], Hiroshi Ogawa[4], Jun Nishioka[1,2]

[1]Pan-Okhotsk Research Center, Institute of Low Temperature Science, Hokkaido University, Sapporo, 060-0819, Japan
[2]Graduate School of Environmental Science, Hokkaido University, Sapporo, 060-0810, Japan
[3]Faculty of Environmental Earth Science, Hokkaido University, Sapporo, 060-0810, Japan
[4]Atmosphere and Ocean Research Institute, The University of Tokyo, Chiba, 277-8564, Japan

*Correspondence to*: Huailin Deng (denghl@lowtem.hokudai.ac.jp), Jun Nishioka (nishioka@lowtem.hokudai.ac.jp)





**Abstract.** Iron (Fe) and macronutrient supplies and their ratios are major factors determining phytoplankton abundance and community composition in the North Pacific. Previous studies have indicated that Okhotsk Sea Intermediate Water and North Pacific Intermediate Water (NPIW) transport sedimentary Fe to the western subarctic Pacific. Although the supply of Fe and macronutrients from subsurface waters is critical for surface phytoplankton productivity, return paths from NPIW to the subsurface and their impact on the abundance and community composition of the organisms have not been fully understood. In this study, Fe and macronutrient turbulent fluxes, as well as the flux ratios from NPIW to surface waters, were calculated based on a chemical dataset, which included Fe and macronutrient concentrations, with turbulent mixing parameters obtained from the same cruise and same station along the 155° E transect in summer. Additionally, vertical flux divergence was calculated from the estimated vertical fluxes. Surface and subsurface phytoplankton community composition was evaluated in the CHEMTAX program based on algal pigment measurements. The results show that diatom abundance is significantly correlated with the vertical fluxes of Fe and macronutrients, especially with Fe and silicate (Si) fluxes, and with the Fe/N flux ratio along the section line. These results suggest that diatom abundance was controlled by Fe supply from subsurface waters in summer. The computed turbulent flux divergence in the subarctic and Kuroshio-Oyashio Transition Area suggests that enhanced concentrations of Fe and Si in the subsurface layer were supplied from NPIW.

## 1 Introduction

Iron (Fe) and nitrate supplies, as well as their ratios, are major factors for determining phytoplankton productivity in the North Pacific subarctic (Tsuda et al., 2003; Takeda, 2011) and subtropical (Rii et al., 2018) gyres, respectively. Diatoms and picocyanobacterium *Prochlorococcus* are dominant phytoplankton groups in the euphotic layer in the North Pacific subarctic (Suzuki et al., 2011; Endo et al., 2018) and subtropical gyres (Yun et al., 2020), respectively. To understand what kind of phytoplankton groups can grow as primary producers in the North Pacific, the factors controlling the supply of Fe and macronutrients, along with the abundance and composition of phytoplankton, need to be studied.

There are two primary Fe sources in surface waters of the oceanic western North Pacific: "aerosol Fe deposition" and "oceanic Fe supply". We need a coherent explanation for the biological response in the western North Pacific waters that incorporates knowledge of both the "aerosol Fe deposition" and the "oceanic Fe supply". Mineral dust and anthropogenic aerosols from East Asia have been regarded as important Fe sources in the western North Pacific (Duce et al., 1991; Jickells et al., 2005; Ito et al., 2019; Kurisu et al., 2021). Although a few reports indicate the dust supply stimulates phytoplankton growth in the subarctic Pacific (e.g., Bishop et al., 2002; Hamme et al., 2010), dust-mediated biological production is still rare in the modern ocean (Boyd et al., 2010). Therefore, other Fe sources, such as oceanic sources, must be considered to explain the major biological activity in the North Pacific (Nagashima et al., 2023; Kurisu et al., 2024).

For the oceanic Fe supply, previous studies indicated that Okhotsk Sea Intermediate Water and North Pacific Intermediate Water (NPIW) carry sedimentary Fe to the western subarctic Pacific, which would further fuel oceanic phytoplankton





productivity (Nishioka et al., 2007, 2013, 2020; Nishioka and Obata, 2017). The influence of NPIW can affect the intermediate water of the subtropical eastern North Pacific (Conway and John, 2015) and the subtropical western North Pacific (Yamashita et al., 2020), which indicates that sedimentary Fe may be widely distributed throughout the North Pacific. Nishioka et al. (2020, 2021) demonstrated that Fe and macronutrients were transported from NPIW to less dense water ($< 26.6$ $\sigma_\theta$) by enhanced high vertical turbulent diapycnal mixing, which was generated by the interaction between tidal currents and rough topography around the marginal sea island chains. They also indicated that Fe might be transported upward from the shallower isopycnal surface of the upper NPIW (26.6 $\sigma_\theta$) to the surface layer by winter surface mixing.

Major macronutrients in subsurface waters can be transported to the surface layer by vertical turbulent diapycnal mixing in the western North Pacific (Rae et al., 2020; Holzer et al., 2021). Sarmiento et al. (2004) utilized a tracer named Si*, which was prepared by combining silicate with nitrate, and concluded that the nutrient return path from the intermediate layer to the surface layer is located in the northwest corner of the Pacific; moreover, NPIW might serve as an important nutrient source which retains the high Si* signature in the subsurface layer in both high- and low-latitude areas. Kaneko et al. (2021) asserted that vertical nitrate transportation occurs from NPIW to the surface layer in the Emperor Seamount with prominent divergence/convergence features, and the nitrate return path might be located in the subtropical Pacific, as internal tides would elevate vertical flux locally. Elevated vertical turbulent nitrate flux was observed in the Kuroshio area, which might imply nitrate transportation from NPIW (Nagai et al., 2019; Tanaka et al., 2019).

To date a few studies have confirmed the relationships between Fe or macronutrient supply fluxes and biological responses. It is noteworthy that the phytoplankton abundance in the western North Pacific vary seasonally (Shimada et al., 2006; Hashioka and Yamanaka, 2007; Nishioka et al., 2011; Takahashi et al., 2013; Nishioka et al., 2021), therefore, we need to understand which Fe supply processes controlling the seasonal variability of the biology. In this study, we mainly focused on the oceanic Fe supply process and quantitatively examined the Fe supply by estimating Fe and macronutrient fluxes from subsurface waters toward the surface and then investigated their relationships with phytoplankton pigment data during summer. In addition, the mechanisms and location of the Fe and macronutrient return path from NPIW to the subsurface layer were investigated.

## 2 Data and methods





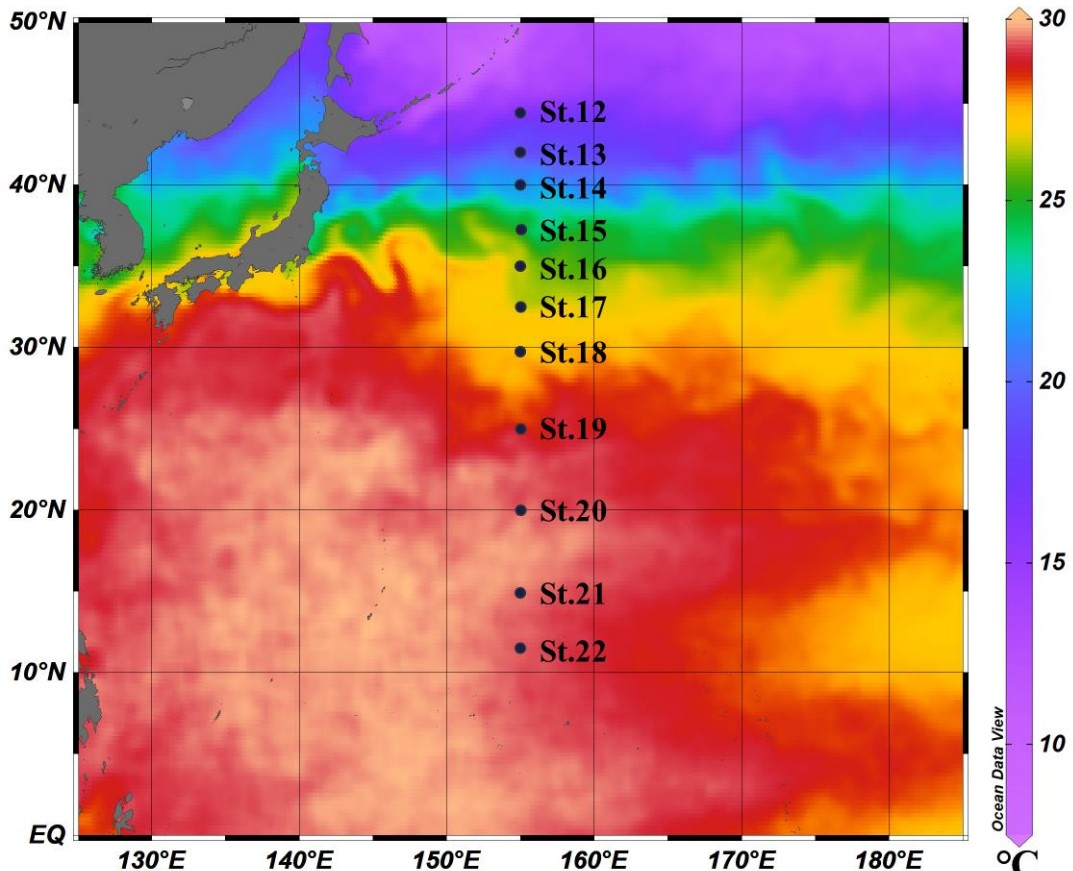


**Figure 1: Location of observed stations from St. 12 to St. 22 in the cruise of KH-08-2. The sea surface temperature (SST) data was cited from NOAA Daily Optimum Interpolation Sea Surface Temperature (OISST) V2.1 High Resolution Dataset (https://www.ncei.noaa.gov/products/optimum-interpolation-sst) (Huang et al., 2021). This figure used the average SST data calculated from August 1st to September 9th, 2008, and was drawn using the software Ocean Data View (ODV; Schlitzer, R., Ocean**
**Data View, http://odv.awi.de, 2016).**

Observation and sample collection were conducted in the western North Pacific during the KH-08-2 expedition of the R/V *Hakuho-maru* (JAMSTEC/ U. Tokyo) from August 1st to September 9th, 2008. All the stations were arranged on a 155° E transect from 40° N to 11.5° N (Fig. 1). For the data used in this study, a chemical dataset was obtained from Nishioka et al.

(2013, 2020), and a physical dataset was obtained from Kaneko et al. (2021).

Hydrography data, including salinity and *in situ* temperature, were measured using a conductivity–temperature–depth (CTD; SBE911plus; SeaBird Electronics Inc.) sensor. Station (hereafter, St.) 12 was located in the subarctic gyre (SAG) between the subarctic front and subarctic boundary. The subarctic front is defined as where the *in situ* temperature is approximately 4 °C

at 100 m depth, and salinity decreases upward (Favorite, 1976). In the subarctic boundary, surface salinity approaches 34.0



(Favorite, 1976). The Kuroshio-Oyashio Transition Area (KOTA) is between the subarctic boundary and the Kuroshio Extension (KE), and the KE is marked as the location at which the temperature is approximately 14 °C at 200 m depth (Kawai, 1969). St.13-15 and St. 16-17 were located in the KOTA and KE, respectively. The subtropical gyre (STG) is south of the KE, where St.18-22 were located.


The chemical dataset used in this study, including dissolved Fe (dFe) and macronutrient (nitrate, phosphate, and silicate) concentrations during the R/V *Hakuho-Maru* KH-08-2 cruise, has already been published (Nishioka et al., 2013, 2020). The chemical dataset includes data on discrete water samples collected from the surface to the bottom (bottom depth 4400−5600 m). The dFe concentrations were determined with a flow-injection analysis (FIA) system (Obata et al., 1993), which was quality-

controlled by the SAFe reference sample (Johnson et al., 2007). Macronutrient concentrations were measured by a BRAN-LUEBBE autoanalyzer (TRACCS 800).

The physical dataset used in this study, including vertical turbulent mixing parameters obtained from the same cruise and same stations as chemical parameter measurements, ranging from the surface to approximately 2000 m depth, has been reported in

Kaneko et al. (2021). The turbulent energy dissipation rate ($\varepsilon$) was measured based on microscale velocity fluctuations derived from a vertical microstructure profiler (VMP2000; Rockland Scientific International Inc.). This method followed the procedure outlined by Itoh et al. (2010), who calculated shear spectra for segments lasting 16 seconds. The integration of these spectra across different wavenumbers yielded the value of $\varepsilon$. The vertical diffusivity ($K\rho$) used in this study was obtained from Kaneko et al. (2021), which was evaluated from the following formula:

$K\rho = \varepsilon\ \Gamma/N^2$, $\qquad\qquad\qquad\qquad\qquad\qquad\qquad\qquad\qquad$ (1)

where the turbulent kinetic energy dissipation rate ($\varepsilon$) was measured by a VMP2000, the mixing efficiency ($\Gamma$) was assumed to be a constant 0.2 (Osborn, 1980), and the square of buoyancy frequency ($N^2$) was calculated as $N^2 = -g/\rho_0\ \partial\sigma_\theta/\partial Z$, where g and $\rho_0$ are the gravitational acceleration and reference potential density, based on Kaneko et al. (2013). As there was large variation in the turbulence, $K\rho$ value was calculated at every 50 m depth to increase the estimation accuracy (Kaneko et al.,

110 2013).

The dFe and macronutrient vertical turbulent flux ($F$) were quantified by the following formula (Kaneko et al., 2021):

$F = -\ K_\rho\ \frac{\partial C}{\partial Z}$ (Lewis et al., 1986) (minus indicated upward flux), $\qquad\qquad\qquad\qquad$ (2)

where $K_\rho$ and $\frac{\partial C}{\partial Z}$ indicate vertical diffusivity and dFe or macronutrient vertical gradient, respectively.


The dFe and macronutrient fluxes are displayed in two ways: 1) flux for the section profile and 2) individual flux (henceforth 'indiv-flux') for the vertical profile at each station. 1) Flux for the section profile represents the vertical flux within the entire section, where this flux was calculated with the vertical gradients of dFe and macronutrient concentrations ($\frac{\partial C}{\partial Z}$) between every



adjacent two discrete water samples depths of the water column, and the vertical diffusivity in the corresponding shallower

layer. 2) For the calculation of the indiv-flux, vertical gradients of dFe and macronutrient concentrations ($\frac{\partial C}{\partial Z}$) were calculated

from the layer where the concentration changed sharply with depth below the surface (calculation details and selected depth

range are shown in Supplement Fig. S1, S2, respectively). The average $K_\rho$ value was used in the same depth range. Also, the

flux ratios (such as the dFe/N flux ratio) were calculated as the ratio of two indiv-fluxes within the same depth range at each

station. Flux for the 1) section profile is used to display the distributions of the fluxes and to calculate the flux divergences; 2)

indiv-flux is used to compare it with biological parameters in each station.

Additionally, vertical flux divergence was calculated from the equation below based on Kaneko et al. (2021):

$$\textit{Flux divergence} = \frac{F_{Z1} - F_{Z2}}{Z_1 - Z_2}, \tag{3}$$

where $Z_1$ and $Z_2$ represent the upper and lower depths of every two fluxes for the section profile (0 m $< Z_1 < Z_2$). $F_{Z1}$ and $F_{Z2}$

were the fluxes for the section profile at the depths of $Z_1$ and $Z_2$.

Divergence occurs when the value of flux divergence is positive, indicating that upward nutrient flux in the upper layer is

larger than in the lower layer. The total nutrient concentration in the corresponding layer decreases where positive divergence

is indicated. On the other hand, convergence occurs when the value of the flux divergence is negative and the total nutrient

concentration in the corresponding layer increases (details are shown in Supplement Fig. S3).

Biological parameters, including chlorophyll $a$ (Chl $a$) and other pigment concentrations, were also measured at the same

stations during the cruise, and the depth range was from the surface to 150 m depth. Chl $a$ concentration was measured onboard

with a Turner Designs fluorometer AU-10 using the non-acidification method of Welschmeyer (1994). The subsurface Chl $a$

maximum (SCM) was evaluated from Chl $a$ data in vertical profiles. The vertically integrated Chl $a$ concentration (VIC: unit

is mg/m$^2$) was calculated based on a trapezoidal rule of the data from the surface to 150 m. Chlorophylls and carotenoids were

also analyzed by high-performance liquid chromatography (HPLC) according to Suzuki et al. (2014, 2021). In brief, duplicate

2.0-2.4 L seawater samples collected with acid-cleaned 12-L Niskin-X bottles were filtered onto Whatman GF/F filters

(nominal pore size 0.7 μm, 25 mm in diameter) under a low-vacuum (<0.013 MPa) pressure. The filtered samples were then

stored in a deep freezer (–80 °C) prior to onshore analysis. The frozen filter was minced and soaked in 3 mL DMF (*N, N*-

dimethyl formamide), and canthaxanthin was added as an internal standard. The pigment extracted by DMF was filtered using

0.45-μm PTFE filters to remove fine particles. A 250 μL mixture of pigment extracts and 1 M ammonium acetate solution

were introduced into an HPLC system (CLASS-VP System, Shimadzu) along with an Agilent ZORBAX Eclipse XDB-C8

Rapid Resolution column (3.5 μm particle size, 4.6 × 150 mm) connected to an Agilent Eclipse XDB-C8 cartridge (3.5 μm

particle size, 4.6 × 30 mm) as the guard column. Based on pigment concentration and composition, phytoplankton community

composition was estimated with the CHEMTAX (version 1.95) program (Mackey et al., 1996). Total Chl $a$ biomass was set



by the sum of HPLC-determined Chl *a* and divinyl Chl *a* (DV Chl *a*) derived from *Prochlorococcus* (Suzuki et al., 1995). For the CHEMTAX analysis, precise estimation requires that the initial ratios are close to those of the assessed phytoplankton assemblage (Latasa, 2007). The initial pigment ratio matrix of Chl *a* for eight taxonomic groups, including diatoms,

dinoflagellates, chlorophytes, prasinophytes, chrysophytes, haptophytes, and cyanobacteria (excluding *Prochlorococcus*), and that of DV Chl *a* for *Prochlorococcus* were selected and modified from Kanayama et al. (2020) during CHEMTAX analysis. The ratio limit was set to 500% to constrain all the pigment calculations (Mackey et al., 1996). To derive the most suitable matrix of phytoplankton community composition, 60 further pigment ratio matrices were introduced (scaling factor = 0.7) based on the initial matrix, and the 6 matrices with the smallest root-mean-square error (RMSE) were selected. The average

matrix of these 6 matrices was regarded as the final matrix (scaling factor = 0.7) and initial matrix for the next run (scaling factor = 0.4). The same procedure was performed, and the second average matrix was chosen for the phytoplankton community composition calculation after comparison (details are shown in Supplement Tables S1-S3).

## 3 Results and discussion

### 3.1 Hydrographic conditions

High concentrations of dFe and macronutrients in NPIW have previously been reported by Nishioka et al. (2013, 2020). Yasuda et al. (2001) proposed that NPIW, with a core density of 26.8 $\sigma_\theta$, can be divided into the upper NPIW (U-NPIW) and the lower NPIW (L-NPIW), which are mainly contributed by Okhotsk Sea water (OSW) and the Western Subarctic Gyre (WSAG) water, respectively. According to Yasuda et al. (2001), the U-NPIW (density range is from 26.6 to 27.0 $\sigma_\theta$) consisted of 71% OSW and 29% WSAG water, whereas the L-NPIW (density range is from 27.0 to 27.5 $\sigma_\theta$) accounted for 75% WSAG and 25% OSW

in the vicinity of the south coast of Hokkaido. Figures 2a and 2b show observed *in situ* temperature and salinity, respectively. The hydrographic data of this study clearly shows the stratification of water masses, indicating the U-NPIW. Particularly in Fig. 2b, the U-NPIW was identified by the vertical salinity minimum (Reid, 1965). In the SAG and KOTA, the 26.5 $\sigma_\theta$ isopycnal surface reaches ~100 m depth and extends to ~600 m in the STG. The low salinity intrusion from the subarctic to the subtropical region is observed in the U-NPIW isopycnal surfaces.




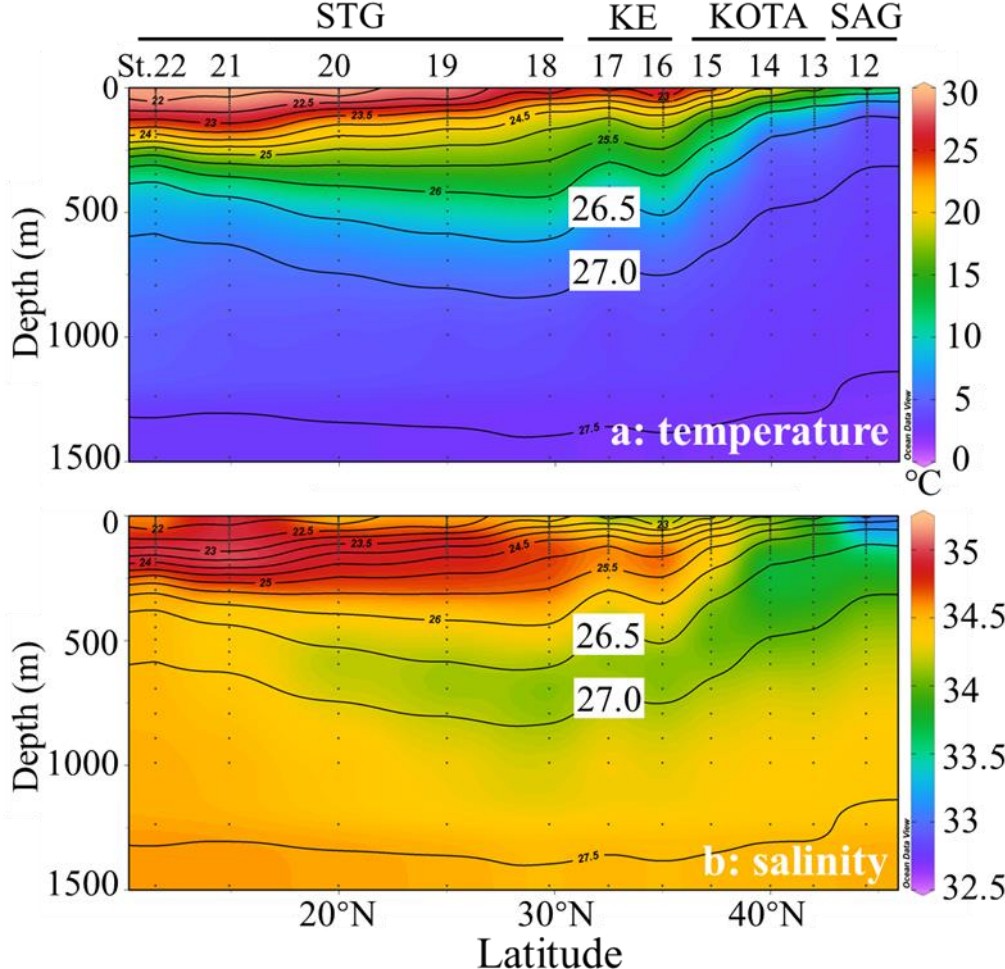

**Figure 2: Section profiles of (a) in situ temperature (°C) and (b) salinity along the observational transect at 155° E from the subarctic gyre to the subtropical gyre. *SAG*: subarctic gyre, *KOTA*: Kuroshio-Oyashio Transition Area, *KE*: Kuroshio extension, *STG*: subtropical area. Black contour lines in the figures denote each potential density ($\sigma_\theta$) with special reference to the 26.5 and 27.0 $\sigma_\theta$ isopycnal surfaces. These figures were drawn using the ODV.**

**3.2 Turbulent mixing**

As shown in Fig. 3, $K_\rho$ varied from $O(10^{-4})$ m$^2$ s$^{-1}$ to $O(10^{-7})$ m$^2$ s$^{-1}$ ($O$ stands for the order of magnitude). $K_\rho$ was high, $O(10^{-5})$ m$^2$ s$^{-1}$ below 26.5 $\sigma_\theta$ in the SAG, KOTA, and KE. In the SAG and KOTA, relatively high $K_\rho$ of $O(10^{-5})$ m$^2$ s$^{-1}$, was observed in the water with a density greater than 26.5 $\sigma_\theta$, which was ~ 200 m at St. 12, 13,14 and was ~ 400 m at St. 15. Whereas, in the STG, $K_\rho$ was not as high and $O(10^{-6})$ m$^2$ s$^{-1}$ between 26.5 $\sigma_\theta$ and 27.0 $\sigma_\theta$, while $K_\rho$ was $O(10^{-5})$ m$^2$ s$^{-1}$ in the density >27.0 $\sigma_\theta$, which was at ~ 600 m depth. In the surface layer, $K_\rho$ was low, $O(10^{-7})$ m$^2$ s$^{-1}$ in the northern part, while high $K_\rho$, $O(10^{-5}\,\text{to}\,10^{-4})$ m$^2$ s$^{-1}$, was observed at St. 20, 21, and 22 in the STG.





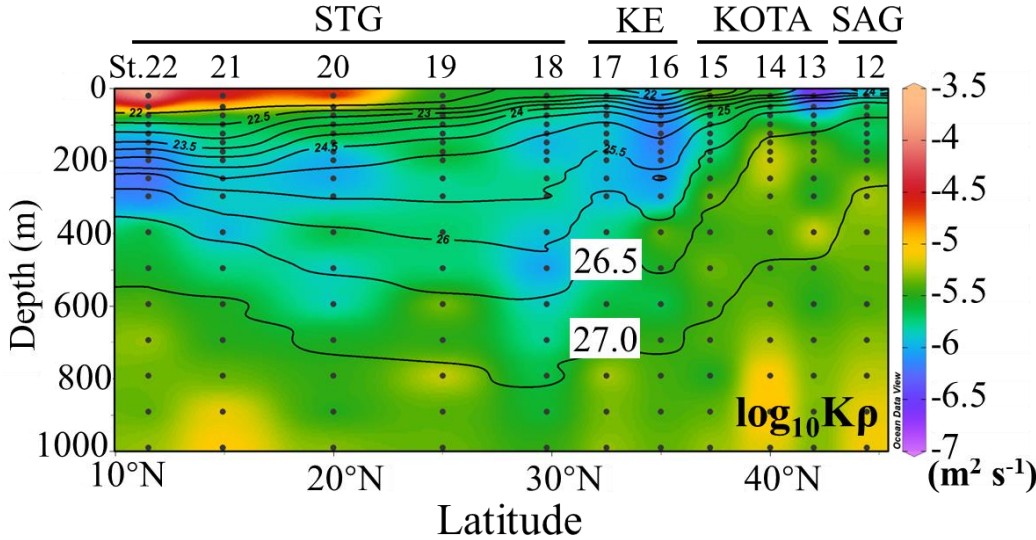


**Figure 3: Vertical section profile of vertical diffusivity ($K\rho$) along the 155° E transect from the subarctic to the subtropical Pacific. Black contour lines in the figure denote each potential density ($\sigma_\theta$) with special attention to the 26.5 and 27.0 $\sigma_\theta$ isopycnal surfaces. This figure was drawn using the ODV.**

### 3.3 Dissolved iron and macronutrient distribution

The vertical section profiles of dFe and macronutrient concentrations along the 155° E transect are shown in Fig. 4. In the SAG, KOTA, and KE, high-dFe water (> 0.6 nM) was observed below the 26.5 $\sigma_\theta$ isopycnal surface and reached ~1 nM below the 27.0 $\sigma_\theta$ isopycnal surface. Particularly in the SAG and the KOTA, high dFe concentrations and steep gradient characteristics were found just below the surface layer (~ 100 m at the SAG station, ~ 200 m in the KOTA). In the STG, high-dFe water and a steep gradient are not observed just below the surface, but a high concentration (> 0.5 nM) is found in the deep layer with a density > 27.0 $\sigma_\theta$.



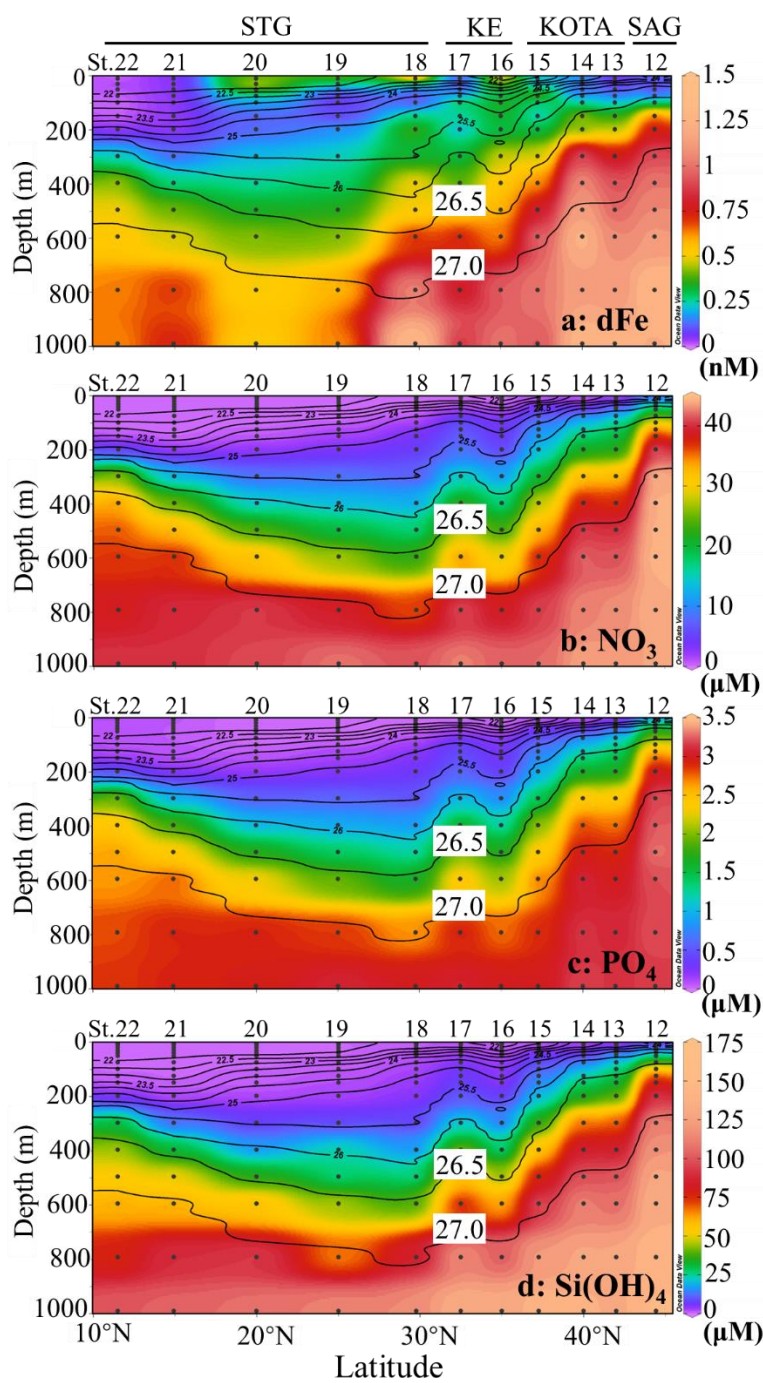

**Figure 4: Vertical section profiles of (a) dissolved Fe, (b) nitrate, (c) phosphate, and (d) silicate concentration along 155° E transect from the subarctic to subtropical Pacific. Black contour lines in the figure denote each potential density (σθ) with special attention to the 26.5 and 27.0 σθ isopycnal surfaces. This figure was drawn using the ODV.**




Note that the macronutrient concentrations follow the isopycnal surface distribution strictly (macronutrient concentrations tend to be uniform along isopycnal surfaces) and increase with density, indicating that the dynamics of macronutrients are controlled by the water mass distribution and circulation. If NPIW reaches low-latitude areas, the macronutrients would also be

transported that far. A high macronutrient concentration core, deeper than 26.5 $\sigma_\theta$, extends to the STG. On the other hand, the high-dFe concentration core in the density >27.0 extends from the SAG to 30° N (St. 18). In contrast to the macronutrient, the high-dFe concentration core does not strictly follow the isopycnal surface distribution. This may be because part of the dFe fraction can be removed by sinking, scavenging, and aggregation during the lateral transport of NPIW (Yamashita et al., 2020; Misumi et al., 2021). Increased dFe is observed near the surface at St. 16, 18, and 20, but macronutrients are deficient. The

result indicates that the surface layer of the KE and STG have a relatively high dFe concentration of ~0.5 nM, which could be caused by the supply from the upstream Kuroshio and/or aerosol Fe input (Kurisu et al., 2021).

**3.4 Dissolved iron and macronutrient flux**

The flux for the section profile was calculated from the adjacent two depths (Fig. 5). In the layer between the 26.0 and 27.0 $\sigma_\theta$ pycnocline, the upward dFe flux remains high, $O(-10^{-8})$ µmol m$^{-2}$ s$^{-1}$, in the SAG and KOTA because both the vertical gradient

of dFe concentration and vertical diffusivity are large. In the same density range (26.0−27.0 $\sigma_\theta$) of the KE and STG, a lower upward dFe flux of $O(-10^{-10})$ µmol m$^{-2}$ s$^{-1}$ is observed because of a smaller dFe concentration gradient and a smaller vertical diffusivity in these regions. Near the surface at St. 20, 21, and 22 in the STG, an elevated upward dFe flux of $O(-10^{-7})$ µmol m$^{-2}$ s$^{-1}$ is observed mainly because of the high vertical diffusivity. Additionally, in the surface or subsurface layers at St. 15, 18, 19, and 20, the downward dFe flux of $O(10^{-9}-10^{-7})$ µmol m$^{-2}$ s$^{-1}$ is seen because of the higher dFe concentration at

shallower depths (Fig. 4a).

In the water with a density lower than 27.0 $\sigma_\theta$ in the SAG and KOTA, a high upward nitrate flux of $O(-10^{-7})$ mmol m$^{-2}$ s$^{-1}$ is observed. Between the 25.5 and 27.0 $\sigma_\theta$ isopycnal surface in the KE and STG, a relatively high nitrate flux of $O(-10^{-8}--10^{-7})$ mmol m$^{-2}$ s$^{-1}$ is observed. In the surface layer of the STG, nitrate flux is seldom detected because the nitrate concentration is

under the detection limit. The phosphate flux displays a similar pattern as the nitrate flux. However, there are distinct patterns between the nitrate and silicate flux in the water with a density greater than 27.0 $\sigma_\theta$; higher upward silicate flux is observed, while nitrate flux is downward.





**Figure 5: Vertical section profile of (a) dissolved Fe flux, (b) nitrate flux, (c) phosphate flux, and (d) silicate flux along 155° E transect from the subarctic to subtropical Pacific. Black contour lines in the figure denote each potential density (σ_θ) with special attention to the 25.5 or 26.0 and 27.0 σ_θ isopycnal surfaces. Upward-directed flux values are negative. This figure was drawn using the ODV.**



**3.5 Relationship between Chl *a* concentration and individual flux or flux ratio**

Kaneko et al. (2021) compared sensor-based Chl *a* concentration (SCM and VIC) with nitrate flux calculated at the bottom of

the euphotic zone at the same stations during the same cruise as in the present study. Consequently, Kaneko et al. (2021) concluded that there was no clear relationship between the nitrate flux and Chl *a* concentration.

In this study, the Chl *a* concentration (SCM and VIC) determined by HPLC were compared with dFe, nitrate, phosphate, silicate indiv-fluxes, and flux ratios. As a result, there is nearly no correlation between total Chl *a* concentration and individual

fluxes or flux ratios (Supplement Fig. S4-S6), and these results are similar to the results by Kaneko et al. (2021).

**3.6 Relationship between Chl *a* concentration from each phytoplankton group and individual flux or flux ratio**

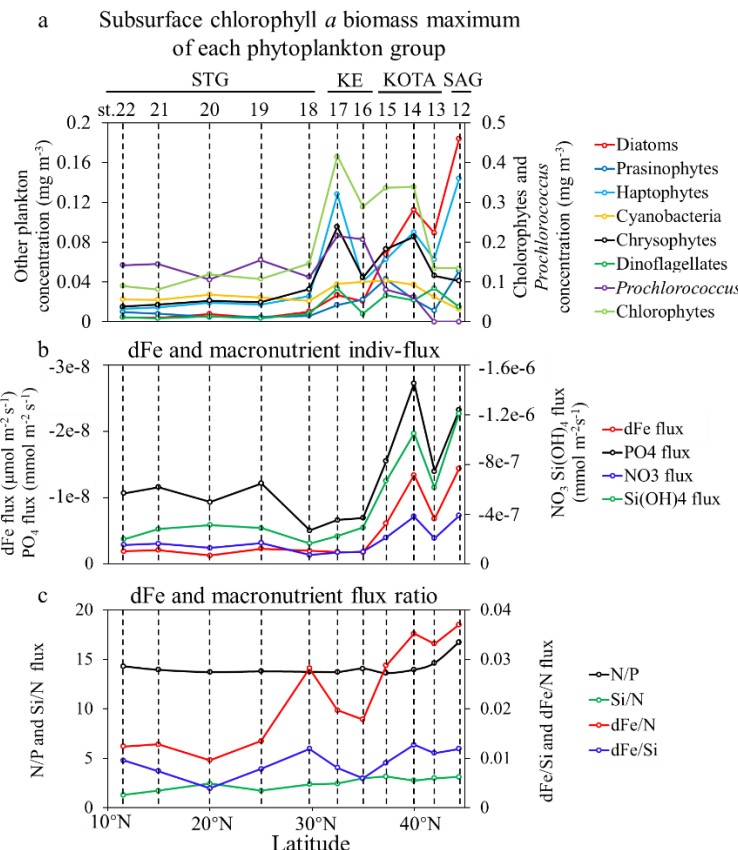

**Figure 6: Latitudinal distribution patterns of (a) subsurface chlorophyll *a* biomass maximum (mg/m³) of phytoplankton groups,**
**including diatoms, prasinophytes, haptophytes, cyanobacteria (except *Prochlorococcus*), chrysophytes, dinoflagellates,**



***Prochlorococcus*, and chlorophytes in terms of chlorophyll *a* (Chl *a*) biomass (b) dissolved Fe and macronutrient indivi-fluxes and (c) dissolved Fe and macronutrient flux ratio.**

The subsurface Chl *a* biomass maximum of each phytoplankton group along the 155° E was estimated with the CHEMTAX

program using algal pigment data, as shown in Fig. 6a. Fig. 6b illustrates dFe and macronutrient indiv-fluxes. In the SAG and KOTA, the dFe and macronutrient indiv-fluxes are higher than those in the KE and STG. However, as shown in Fig. 6c, the dFe/N flux ratio is higher in the SAG and KOTA than in the KE and STG. Little difference is found in the other flux ratios among regions.

The distribution patterns of diatom-derived Chl *a* concentration and the subsurface diatom-derived Chl *a* maximum (SDM) are shown in Fig. 7a and 7b, respectively. Higher diatom-derived Chl *a* concentrations (~ 0.1 mg m$^{-3}$) are observed in the SAG above 100 m depth. In general, the SDM increases with latitude. The results of diatom distribution along 155° E in this study are similar to those of previous studies. The abundance of diatoms, based on the 18SrRNA gene copy numbers, along 160° E in summer reached the maximum in the SAG (Endo et al., 2018). The multi-annual (2002–2022) averaged diatom-derived Chl

*a* concentration also increased with latitude in the western North Pacific in summer (June–August), based on satellite ocean color remote sensing (Li et al., 2023).

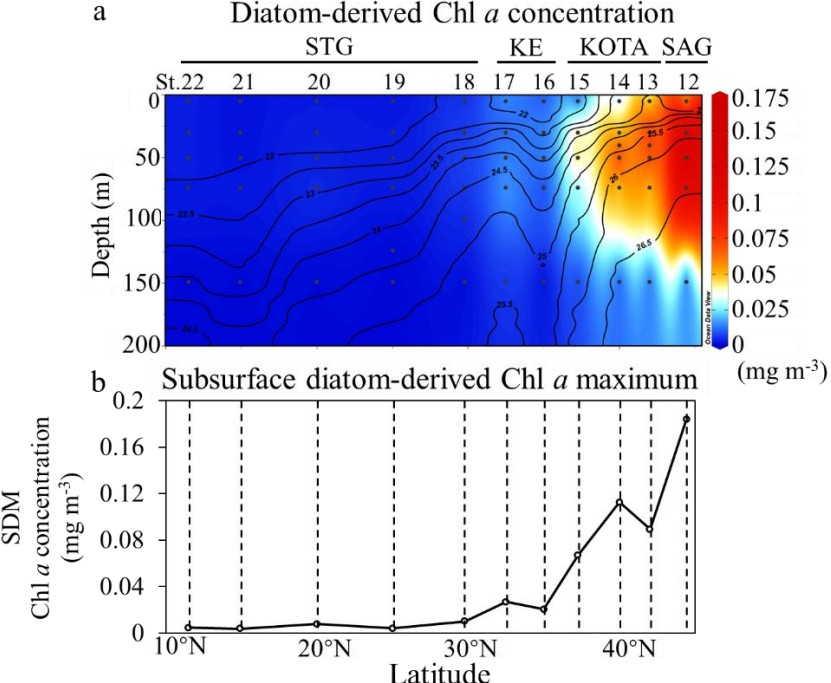

**Figure 7: Section profiles of (a) diatom-derived Chl *a* concentration (mg m$^{-3}$), (b) the subsurface diatom-derived Chl *a* maximum**

**(SDM) concentration (mg m$^{-3}$) at each station. Figure 7a was drawn using the ODV.**





The SDM can be observed at the depth range of 30–75 m at all stations in this study. The depth range of indiv-fluxes, which indicated the nutrient supply from the subsurface, included the depth of SDM in the SAG, KOTA, KE, and some stations in the STG (St. 12–19). In the rest stations in the STG, indiv-fluxes were calculated just below the depth of SDM, from ~200 m

depth at St. 20–21 and from ~150 m depth at St. 22. Pearson correlation analysis was employed to explore the relationships between dFe, macronutrient fluxes, and phytoplankton abundance, with the coefficient of determination ($r^2$) and $p$-value. As a result, linear correlations were found between the SDM and dFe or macronutrient indiv-fluxes (Fig. 8a–8d). These results indicate that the diatom abundance is highly controlled by dFe and macronutrient supply from the subsurface below. Based on the values of $r^2$ and $p$-value, dFe and Si more strongly influence the diatom-derived Chl $a$ concentration than nitrate and

phosphate. For the flux ratios, the dFe/N flux ratio is correlated with the diatom-derived Chl $a$ concentration (Fig. 8e–8h). Previous studies have proven that diatom growth was often limited by nutrient supply (Allen et al., 2005), including dFe (Hogle et al., 2018; Nishioka et al., 2021). Martin and Fitzwater (1988) demonstrated that Fe addition would stimulate the phytoplankton growth in the high nutrient-low chlorophyll (HNLC) area through an on-deck incubation experiment. Diatom abundance was regulated by Fe bioavailability in the subarctic North Pacific through the mesoscale iron fertilization

experiment (Tsuda et al., 2003). Different from other phytoplankton communities, diatoms own the silicified frustules that protect them against predators (Hamm et al., 2003), and diatoms with high nutrient uptake and high growth rate characteristics would utilize nutrients quickly for enhanced growth (Sommer, 1984; Litchman et al., 2007). As a result, dFe and macronutrient fluxes from subsurface waters control diatom abundance, especially dFe flux in summer.

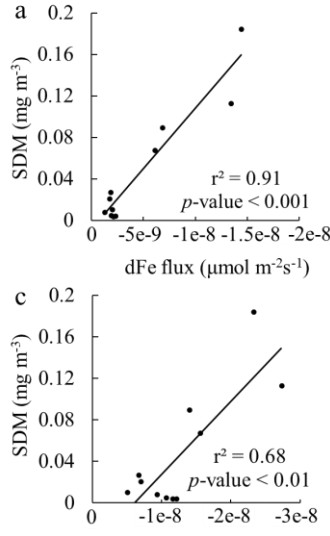

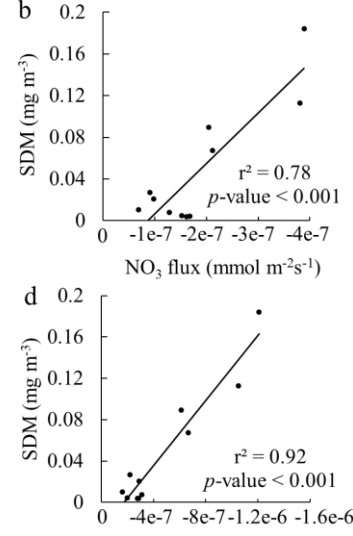





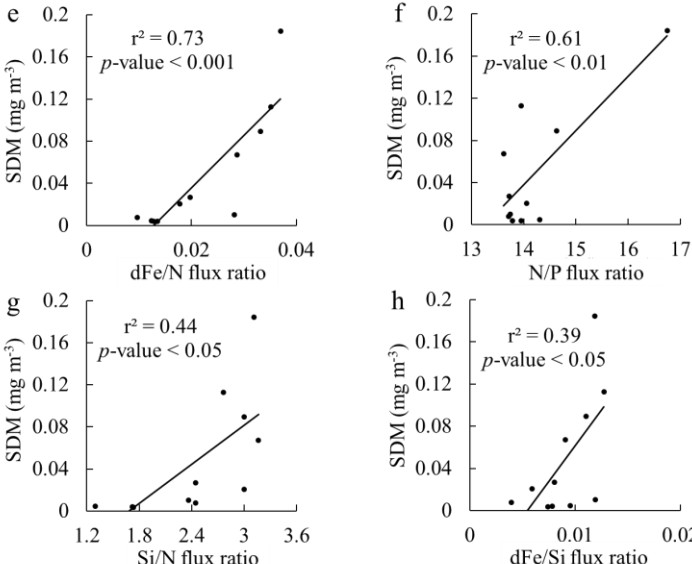

**Figure 8: Comparisons of the subsurface diatom-derived Chl *a* maximum (SDM) to indiv-fluxes or flux ratios. Plots of the SDMvs. (a) dissolved Fe, (b) nitrate, (c) phosphate, and (d) silicate indiv-fluxes. Plots of SDM vs. (e) dFe/nitrate, (f) nitrate/phosphate, (g) silicate/nitrate, and (h) dFe/silicate flux ratios. The results of linear regression analysis with the coefficient of determination ($r^2$) and**
***p*-values are shown in each plot.**

The Pearson correlation coefficient (r) between the subsurface Chl *a* maximum derived from each phytoplankton species and

dFe or macronutrient indiv-fluxes (shown in Fig. 9) indicates that diatoms, followed by haptophytes, prasinophytes are highly

positively correlated, and *Prochlorococcus* is negatively correlated, with dFe and macronutrient indiv-fluxes from subsurface

waters (r > 0.4).

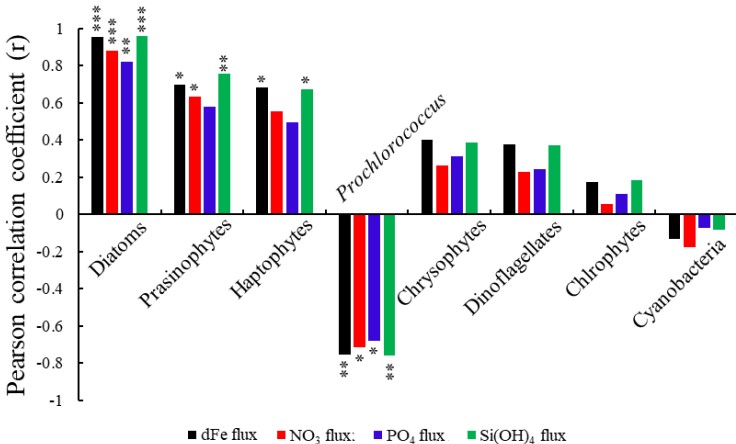





**Figure 9: The Pearson correlation coefficient (r) between the subsurface Chl *a* maximum concentration of each phytoplankton group, dissolved Fe, and macronutrient indiv-fluxes (\*\*\**p*-value <0.001, \*\**p*-value <0.01, \**p*-value <0.05). Note that *Prochlorococcus* and other cyanobacteria are negatively correlated to indiv-fluxes, whereas other phytoplankton groups are positively correlated with indiv-fluxes.**


Previous studies indicated that haptophytes dominated phytoplankton communities in the subtropical and subarctic North Pacific (Hirata et al., 2011), especially in the Kuroshio ecosystems (Endo and Suzuki, 2019). According to Wang et al. (2022),

haptophyte abundance in the KOTA increased in proportion to nutrient supply and was markedly high in the KE in the depth of subsurface chlorophyll maximum, which is similar to the haptophyte distribution in this study, as shown in Fig. 6a. The haptophyte cell abundances were higher in the Fe addition treatments than those in the control treatments through the incubation studies (Endo et al., 2017). Nitrate addition stimulated the carbon fixation rates of haptophytes in nitrate-depleted waters (Mills et al., 2020). The haptophyte community composition was positively correlated with nitrate plus nitrite and

phosphate concentrations by redundancy analysis with seawater samples collected across the Tokara Strait (Endo and Suzuki, 2019). In the haptophyte *Hyalolithus neolepis* occurring in the subarctic western Pacific, one scale type of the non-motile form undergoes intracellular silicification (Yoshida et al., 2006), suggesting a silica requirement. These studies coincide with the result of this study, which is that haptophyte abundance is controlled by the dFe and macronutrient fluxes from the subsurface.

Prasinophytes were predominant in the phytoplankton assemblage in the western SAG during summer (Suzuki et al., 2002). During the Subarctic Pacific Fe Experiment for Ecosystem Dynamics Study (SEEDS II), prasinophytes were predominant in the phytoplankton communities in the Fe-fertilized patch in the western North subarctic Pacific (Suzuki et al., 2009). A numerical modelling study indicated that prasinophytes were limited by Fe supply in the HNLC region of the subarctic Pacific (Litchman et al., 2006). Nitrate and phosphate supply stimulated the growth of prasinophyte *Micromonas* sp. and *Tetraselmis*

*suecica* (a genus of prasinophytes), respectively, through the incubation experiment (Laws et al., 2011; Liefer et al., 2018). These studies indicate that prasinophyte abundance is limited by Fe and macronutrient availability, which agrees with this study that prasinophyte abundance is impacted by the dFe and macronutrient fluxes from the subsurface.

Fig. 9 indicates the negative correlations between *Prochlorococcus* abundance and nutrient fluxes. The negatively correlated

results indicated that the *Prochlorococcus* abundance may be controlled by other factors. In Fig. 6a, *Prochlorococcus* is predominant in the STG and rare in the SAG. Previous studies indicated that *Prochlorococcus* abundance increases with temperature because of their higher metabolic rates (Sarmiento and Gruber, 2006). *Prochlorococcus* cells are seldom observed where the temperature is below 15 °C (Johnson et al., 2006). Also, high *Prochlorococcus* abundance was accompanied by low nutrient supply in subtropical regions (Zubkov et al., 2000). Yun et al. (2020) also reported that *Prochlorococcus* dominated

the phytoplankton biomass in the surface and subsurface layers of the Philippine Sea, where a weak nutrient supply occurred. Our results suggest that *Prochlorococcus* is not controlled by dFe and macronutrient fluxes; a higher abundance of




*Prochlorococcus* can be observed in higher water temperatures and lower nutrient supply areas, like the STG. The temperature is an important factor for controlling this species, as reported in previous studies, rather than dFe and macronutrient fluxes.

### 3.7 Transport of dissolved iron and macronutrients from NPIW to the subsurface layer

The dFe flux divergence indicates that water with a density between 26.6 and 27.1 $\sigma_\theta$ has divergence characteristics, and water with a density lower than 26.6 $\sigma_\theta$ has convergence characteristics in the SAG and KOTA. The divergence characteristics indicate a decreasing trend of dFe concentration in the U-NPIW, and the convergence characteristics indicate an increasing trend of dFe concentration in the water above the U-NPIW in the SAG and KOTA. This process suggests a tendency that dFe is supplemented from the U-NPIW to the water above the 26.6 $\sigma_\theta$ isopycnal surface in the SAG and KOTA. Therefore, a

relatively low concentration of dFe (Fig. 4a) in the surface layer of the SAG and KOTA implies that dFe would be consumed by phytoplankton in the near-surface water.



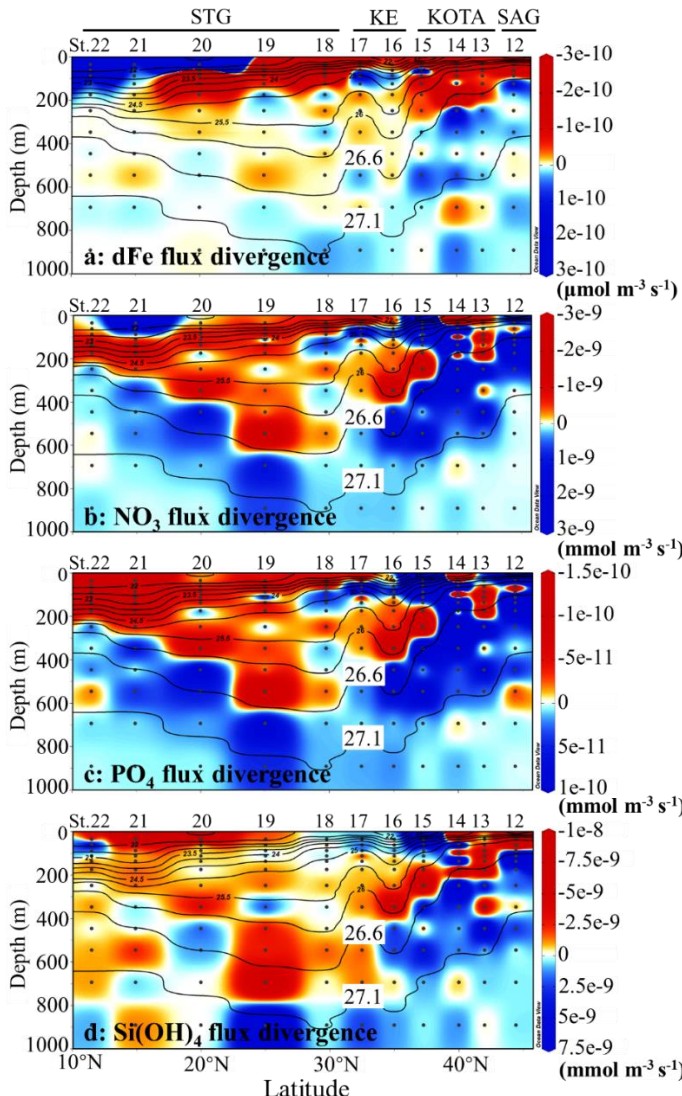

**Figure 10: Vertical section profiles of (a) dissolved Fe flux divergence (μmol m$^{-3}$ s$^{-1}$) along 155° E transect from the subarctic to**
**subtropical Pacific, (b) nitrate flux divergence (mmol m$^{-3}$ s$^{-1}$), (c) phosphate flux divergence (mmol m$^{-3}$ s$^{-1}$) and (d) silicate flux divergence (mmol m$^{-3}$ s$^{-1}$). The red and blue colors indicate the convergence and divergence of each nutrient flux, respectively. The highlighted 26.6 and 27.1 $\sigma_\theta$ isopycnal surfaces suggest the shallower and deeper boundaries of the upper NPIW (U-NPIW). These figures were drawn using the ODV.**

The flux divergences of macronutrients (nitrate, phosphate, and silicate) in the SAG and the KOTA show a divergence

pattern between 26.6 and 27.1 $\sigma_\theta$ (the U-NPIW). The silicate flux divergence indicates that water above the 26.6 $\sigma_\theta$ isopycnal

surface in the SAG and northern part of KOTA at St. 12 and 13 shows a convergence pattern (Fig. 10d). This Si flux

convergence tends to increase Si concentration in the surface layer in the similar manner to the dFe pattern in St. 12 and 13.

This process indicates that Si is transported from the U-NPIW to the water above 26.6 $\sigma_\theta$ in the SAG and northern part of



KOTA at St. 12 and 13. For nitrate and phosphate flux divergence, convergence characteristics can be captured in the water above 26.6 $\sigma_\theta$ in the northern part of KOTA at St. 13 (Fig. 10b, c).

## 4 Conclusions

In previous studies, phytoplankton abundance was confirmed to be influenced by physical properties, such as light, temperature, and chemical properties, including macronutrient concentration. This study firstly demonstrated strong correlations between
dFe or macronutrient fluxes and species-specific Chl *a* concentration, especially for diatoms, in the western North Pacific in summer. The results have revealed that phytoplankton abundance and community composition in the western North Pacific in summer were strongly controlled by dFe and macronutrient fluxes from the subsurface and its stoichiometry. The diatoms were highly regulated by dFe and Si flux from the intermediate water to the surface, as well as the dFe/N flux ratio. Similar to diatoms, haptophytes, and prasinophytes were moderately controlled by dFe and macronutrient fluxes. Fluxes of dFe and
macronutrients from the subsurface layers have affected little other phytoplankton communities. Based on the flux divergence estimates, we conclude that the vertical turbulent transport of dFe and silicate from NPIW to the subsurface layer occurs in the subarctic gyre and Kuroshio-Oyashio Transition Area of the western North Pacific, and the dFe and macronutrient supplies significantly control diatom abundance. Our findings contribute to understanding oceanic biogeochemical circulation in the North Pacific.

## Data availability

The chemical data, including dissolved iron, nitrate, phosphate, and silicate concentration used to produce the figures in this paper, are available from Nishioka et al. (2020) (https://eprints.lib.hokudai.ac.jp/dspace/handle/2115/77482). The physical data, including vertical diffusivity, are publicly available online from Kaneko et al. (2021) (https://ocg.aori.u-tokyo.ac.jp/omix/Kaneko_etal_2020JO/). The biological and CTD data associated with this publication will be deposited in
the data archive site of Hokkaido University when the manuscript is accepted.

## Author contributions

Each named author has participated sufficiently in the work to take public responsibility for the content. JN, IY, KS developed the concept. JN, KS, IY and HO collected the samples and data. JN, KS and HO analysed the samples. HD and JN analysis the dataset and drafted the manuscript. KS, IY and HO reviewed and edited the manuscript.

## Competing interests

Koji Suzuki is a member of the editorial board of Biogeosciences.



**Acknowledgements**

This cruise, identified as KH-08-2, was supported by the Cooperative Research Program of the Atmosphere and Ocean Research Institute, The University of Tokyo. We thank the crew of the R/V Hakuho-maru for their cooperation and assistance
on board.

**Financial support**

This work was supported by the Ministry of Education, Science, and Culture KAKEN grants B "22H03728 and 23H03516", KAKEN grant A "21H04921" and KAKEN grant S "21H05056". This work was supported by a Grant-in-Aid for Scientific Research "22H05205, 20H05598, 20H04965" and the Grant for Joint Research Program of the Institute of Low Temperature
Science, Hokkaido University. The present work was also supported by JST SPRING "JPMJSP2119" and CREST "JPMJCR23J4".

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
