# Peer review of "Phytoplankton community structure in relation to iron and macronutrient fluxes from subsurface waters in the western North Pacific during summer"

_EGUsphere, 2024_

## Author Comment (AC1)

**Hokkaido University**
Institute of Low Temperature Science

**General comments**

This manuscript investigates the phytoplankton community species composition response to iron and macronutrient in the western North Pacific in summer by looking at physically driven inputs. Results show that the response of the diatom community was driven by the vertical fluxes of Fe and macronutrients (especially silicate) being supplied from the North Pacific Intermediate Water. Overall, the results are relevant to the field, and this paper should be published; however, some revisions are required. Currently, the figures would need a bit of work to increase the quality. The discussion on phytoplankton genus-specific response lacks a bit of literature and is not contextualised enough.

Thanks very much for all the comments and evaluation of our manuscript. We appreciate your recognition of the relevance of our study to the field and your suggestions to improve the quality of our figures and strengthen the discussion on the phytoplankton responses at the class level. All comments are very useful in improving the manuscript. All the comments below have been incorporated into our revised manuscript according to the reviewers' suggestions. We hope the revised manuscript will satisfactorily meet your expectations and address your concerns.

As suggested in your general comments, we have improved the discussion of the responses of diatoms to iron availability and the resolution of figures. Please review the point-by-point responses to the specific and technical comments of Reviewer#1-7 to Reviewer#1-10.

**Specific and Technical comments**

Reviewer#1-1
line 52: It is unclear how Si* was derived in Sarmiento et al. 2004 method the way it is written now. In the paper, they use the following: Si*= [Si(OH)4] -[NO3 2].

[Figure]

[Figure]

Thank you for the suggestion. We have modified the sentence in the revised manuscript as follows:

"*Sarmiento et al. (2004) utilized a tracer named Si\* (Si\* = Si(OH)$_4$ – NO$_3^-$), which was defined by combining silicate with nitrate…*".

Reviewer#1-2
Line 63: verb missing 'processes ARE controlling'

Thank you for pointing it out. We have amended it in the revised manuscript as follows:

"*processes controlling*" -> "*processes are controlling*".

Reviewer#1-3
Line 66 and 67: repetition of 'investigated'

Thank you for the suggestion. We have amended it in the revised manuscript as follows:

"*investigated*" -> "*evaluated*".

Reviewer#1-4
Paragraph 3.5 - both parts could be better connected (previous work from Kaneko et al. 2012 and this one), and how this is surprising could be discussed.

Thank you for the suggestion. We combined the two paragraphs to explain the differences between Kaneko et al. (2021) and this study, and added one more sentence to the end of Section 3.5 in the revised manuscript as follows:

"*However, as the dominant phytoplankton communities would vary between the subarctic and subtropical Pacific, relationships between the phytoplankton community composition (cf. Chl a in Kaneko et al. (2021)) and dFe or macronutrient fluxes would attract more attention.*"

[Figure]

[Figure]
**Hokkaido University**
Institute of Low Temperature Science

Reviewer#1-5
Line 262-263: which previous studies? Authors could give examples to contextualise the results. After discussing genetic and remote sensing studies, authors could first compare with in situ data (even from other locations) and then contextualise with remote sensing data.

Thank you for the suggestions. We have added the past literature and modified the sentences in the revised manuscript as follows:

"*The results of diatom distribution along 155° E in this study are similar to those of previous studies (Hirata et al., 2011; Endo et al., 2018; Li et al., 2023), which can provide more insights on the ecology of the organisms. Diatom abundance, as estimated from the 18S rRNA gene copy numbers, reached the maximum in the SAG and showed similar distribution patterns along 160° E in summer (Endo et al., 2018). Additionally, satellite-based diatom-derived Chl a concentrations reported by Hirata et al. (2011) and Li et al. (2023) also increased with latitude in the western North Pacific. These studies would help contextualize our findings within the regional diatom dynamics.*"

Reviewer#1-6
Line 282-285: Martin's work was the first to report that, but since then, numerous Fe-addition bottle incubation experiments were performed- As well as bioavailable Fe assays were performed in the SO; this could be added to contextualise this part of the discussion better.

Thank you for the suggestion. We have added the past literature and modified sentences in the revised manuscript as follows:

"*Since then, a number of ocean iron fertilization experiments (Coale et al., 1996; Boyd et al., 2000; Tsuda et al., 2003; Boyd et al., 2004; Coale et al., 2004), Fe-addition bottle incubation experiments (Martin et al., 1990; Hutchins and Bruland, 1998; Nishioka et al., 2009) and bioavailable Fe assays (Nodwell and Price, 2001; Hassler and Schoemann, 2009), these studies proved that Fe stimulated the diatom growth.*"

Reviewer#1-7

Line 285: Here, the genus-specific requirements towards trace metals and macronutrients could be added; why do diatoms may have higher Fe requirements compared to the other members of the phytoplankton community in the area?

Thank you for the comments. We have added a sentence to the revised manuscript as follows:

*"experiment (Tsuda et al., 2003). Compared to other phytoplankton groups, the requirement of Fe and macronutrients in diatom is higher due to their larger cell size (Sunda and Huntsman, 1995). Also, the Fe starvation–induced protein 1 (ISIP1), which evolves in the Fe uptake process, is primarily a specific protein for diatoms (Kazamia et al., 2018)."*

Reviewer#1-8
288: As it ends with the seasonality aspect, the authors could hint at other controls that could happen during winter, for instance.

Thank you for reviewer's suggestion. We added a sentence to the revised version as follows:

"*Contrasted to the winter season when winter deep mixing brings nutrients from the subsurface to the surface layer, eddy diffusion is one of the important physical processes that supply the nutrients from the subsurface to the surface in summer (Itoh et al., 2021).*"

Reviewer#1-9
It is a bit confusing to have the part 3.7 at the end as the genus-specific response is discussed and the physical drivers were treated before. Maybe authors could consider having in part 1: the physical drivers (incl. fluxes and flux divergence) and then the phytoplankton responses.

Thank you for the suggestion. Below are our explanations on this:

[Figure]

Firstly, we wondered whether the dFe and macronutrient fluxes from subsurface waters to the surface layer could control the phytoplankton abundance. If there is no significant correlation between them, we consider investigating the nutrient return path meaningless.

Since we found significant relationships between them, we started to investigate the location of the return path of dFe and macronutrients from the subsurface to the surface layers via the flux divergence estimates.

Reviewer#1-10
Fig. 6, 7 - the quality is not so good, especially when zooming. Would need more space between the 3 panels to see the titles better.

Thank you for the suggestion. We have taken the following steps to address your concerns:
1. We have recreated Figures 6 and 7 with higher resolutions to ensure better quality;
2. We have added more space between the three panels.

**References**

Boyd, P.W., Law, C.S., Wong, C.S., Nojiri, Y., Tsuda, A., Levasseur, M., Takeda, S., Rivkin, R., Harrison, P.J., Strzepek, R. and Gower, J.: The decline and fate of an iron-induced subarctic phytoplankton bloom, Nature, 428, 549-553, https://doi.org/10.1038/nature02437, 2004.

Boyd, P.W., Watson, A.J., Law, C.S., Abraham, E.R., Trull, T., Murdoch, R., Bakker, D.C., Bowie, A.R., Buesseler, K.O., Chang, H. and Charette, M.: A mesoscale phytoplankton bloom in the polar Southern Ocean stimulated by iron fertilization, Nature, 407, 695-702, https://doi.org/10.1038/35037500, 2000.

Coale, K.H., Johnson, K.S., Chavez, F.P., Buesseler, K.O., Barber, R.T., Brzezinski, M.A., Cochlan, W.P., Millero, F.J., Falkowski, P.G., Bauer, J.E. and Wanninkhof, R.H.: Southern Ocean iron enrichment experiment: carbon cycling in high-and low-Si waters, Science, 304, 408-414, https://doi.org/10.1126/science.1089778, 2004.

[Figure]

[Figure]
 Hokkaido University
Institute of Low Temperature Science

Coale, K.H., Johnson, K.S., Fitzwater, S.E., Gordon, R.M., Tanner, S., Chavez, F.P., Ferioli, L., Sakamoto, C., Rogers, P., Millero, F. and Steinberg, P.: A massive phytoplankton bloom induced by an ecosystem-scale iron fertilization experiment in the equatorial Pacific Ocean, Nature, 383, 495-501, https://doi.org/10.1038/383495a0, 1996.

Hassler, C. S. and Schoemann, V.: Bioavailability of organically bound Fe to model phytoplankton of the Southern Ocean, Biogeosciences, 6, 2281–2296, https://doi.org/10.5194/bg-6-2281-2009, 2009.

Hirata, T., Hardman-Mountford, N. J., Brewin, R. J. W., Aiken, J., Barlow, R., Suzuki, K., Isada, T., Howell, E., Hashioka, T., Noguchi-Aita, M., and Yamanaka, Y.: Synoptic relationships between surface Chlorophyll-a and diagnostic pigments specific to phytoplankton functional types, Biogeosciences, 8, 311–327, https://doi.org/10.5194/bg-8-311-2011, 2011.

Hutchins, D.A. and Bruland, K.W.: Iron-limited diatom growth and Si: N uptake ratios in a coastal upwelling regime, Nature, 393, 561-564, https://doi.org/10.1038/31203, 1998.

Itoh, S., Kaneko, H., Kouketsu, S., Okunishi, T., Tsutsumi, E., Ogawa, H. and Yasuda, I.: Vertical eddy diffusivity in the subsurface pycnocline across the Pacific, J. Oceanogr., 77, 185-197, https://doi.org/10.1007/s10872-020-00589-9, 2021.

Kazamia, E., Sutak, R., Paz-Yepes, J., Dorrell, R.G., Vieira, F.R.J., Mach, J., Morrissey, J., Leon, S., Lam, F., Pelletier, E. and Camadro, J.M.: Endocytosis-mediated siderophore uptake as a strategy for Fe acquisition in diatoms, Sci. Adv., 4, p.eaar4536, https://doi.org/10.1126/sciadv.aar4536, 2018.

Martin, J.H., Fitzwater, S.E. and Gordon, R.M.: Iron deficiency limits phytoplankton growth in Antarctic waters, Global Biogeochem. Cy., 4, 5-12, https://doi.org/10.1029/GB004i001p00005, 1990.

Nishioka, J., Takeda, S., Kondo, Y., Obata, H., Doi, T., Tsumune, D., Wong, C.S., Johnson, W.K., Sutherland, N. and Tsuda, A.: Changes in iron concentrations and bio-availability during an open-ocean mesoscale iron enrichment in the western subarctic

[Figure]

[Figure]

Hokkaido University
Institute of Low Temperature Science

Pacific, SEEDS II, Deep-sea Res. Pt. II, 56, 2796-2809, https://doi.org/10.1016/j.dsr2.2009.06.006, 2009.

Nodwell, L.M. and Price, N.M.: Direct use of inorganic colloidal iron by marine mixotrophic phytoplankton, Limnol. Oceanogr., 46, 765-777, https://doi.org/10.4319/lo.2001.46.4.0765, 2001.

Sunda, W.G. and Huntsman, S.A.: Iron uptake and growth limitation in oceanic and coastal phytoplankton, Mar. Chem., 50, 189-206, https://doi.org/10.1016/0304-4203(95)00035-P , 1995.

Tsuda, A., Takeda, S., Saito, H., Nishioka, J., Nojiri, Y., Kudo, I., Kiyosawa, H., Shiomoto, A., Imai, K., Ono, T. and Shimamoto, A.: A mesoscale iron enrichment in the western subarctic Pacific induces a large centric diatom bloom, Science, 300, 958-961, https://doi.org/10.1126/science.1082000, 2003.

End

---

## Author Comment (AC2)

**Hokkaido University**
Institute of Low Temperature Science

General comments

In this manuscript, Deng et al. take advantage of existing datasets: discrete bottle measurements and vertical microscale velocity fluxuations, to derive diffusive flux of the essential nutrients, iron, nitrate, phosphate, and silicate across an oceanographic transect which extends from the subarctic gyre to the subtropical gyre. The authors relate the derived diffusive fluxes with phytoplankton abundance, separated by taxa using pigment HPLC and CHEMTAX analysis. This work stands to contribute to our understanding of how nutrients and water mass mixing select for specific taxa during the productive summer season. It is important to relate rates of nutrient supply and their ratios to phytoplankton as this would be expected to relate directly to biological metrics, such as growth.

Thank you very much for your positive and constructive feedback on our manuscript. We appreciate your recognition of the relevance of our study to the field.

The manuscript is well-written and constructed in a manner that is easy to follow. I especially like all the figures, they do look almost publication-ready, though could be made slightly larger. I found that while the correlations are the most important contribution (along with derivation of fluxes for this region), the interpretation could be elaborated upon (see listed comments below). I do not think the focus was on phytoplankton responses, but rather relating distributions of specific taxa with vertical fluxes, and the title should reflect this.

Again, thanks very much for the reviewer's comments. We are pleased that you found the manuscript well-written and appreciated the clarity and quality of the figures. As suggested by the reviewer, the title has been changed as follows:

*"Distribution patterns of phytoplankton groups controlling by the iron and macronutrient fluxes from subsurface waters in the western North Pacific during summer".*

[Figure]

[Figure]

**Hokkaido University**
Institute of Low Temperature Science

Specific comments

**Reviewer#2-1**
Line 28: rephrase. I think the main goal is to understand how the environment selects for specific phytoplankton groups.

Thank you for the reviewer's suggestion. We have modified the sentence in the revised manuscript as follows:

"*To understand the factors controlling the supply of Fe and macronutrients, and how the supply would influence the phytoplankton communities, along with the abundance and composition of phytoplankton groups, need to be studied simultaneously.*"

**Reviewer#2-2**
Line 30: "need to be studied, simultaneously."

Thank you for the suggestion. Accordingly, we have added the word "simultaneously" to the revised manuscript.

**Reviewer#2-3**
Line 32: I think you could add a sentence of why it is important to understand Fe supply in this region. For example, distribution of zooplankton, fisheries, higher trophic organisms?

Thank you for the suggestion. The following sentences have been added to the revised manuscript:

"*The availability of Fe and macronutrients can influence the higher trophic organisms through primary producers (i.e., phytoplankton). According to FAO (2020), the western North Pacific accounts for 25% of the global fishery production. Therefore, it is essential to know the mechanisms of Fe and macronutrient supply with the response of phytoplankton communities.*"

**Reviewer#2-4**

[Figure]

[Figure]

Line 38: "A more complete view of Fe biogeochemistry requires oceanic sources also be considered…" I think in this paragraph, it would be good to elaborate a little about why dust deposition doesn't stimulate blooms. Timescale for biological response versus episodic dust deposition events? Bioavailability of this Fe? Not Fe/P limited (N-limited?).

Thank you for the suggestions. We have added the following sentences to the revised manuscript:

"*Although a few reports indicated that the dust supply stimulates phytoplankton growth in the subarctic Pacific (e.g., Bishop et al., 2002; Hamme et al., 2010), and the anthropogenic source of Fe has high bioavailability (Kurisu et al., 2024), dust-mediated biological production enhancement is still rare to be observed in the ocean because of sporadic aerosol Fe supply and scarce sampling opportunities (Boyd et al., 2010). Therefore, oceanic Fe sources must be considered for the major biological production in the North Pacific (Nagashima et al., 2023).*"

Reviewer#2-5
Line 42: I think for non-trace metal chemists, you might want to indicate that sedimentary Fe is derived from the continental shelves. (see also line 45)

Thank you for the suggestion. We have modified the sentences in the revised manuscript as follows:

In line 42,
"*sedimentary Fe derived from the continental shelves to the western subarctic Pacific…*".

In line 45:
"*which indicates that sedimentary Fe derived from the continental shelves could be distributed widely*".

Reviewer#2-6
Line 49: "winter surface mixing" do you mean deep convective mixing?

Yes, you are right. We have changed *"winter surface mixing"* to *"winter deep convective mixing"* in the revised manuscript to avoid confusion.

Reviewer#2-7
Line 53: please put the equation for Si* or be more specific about how silicate and nitrate concentrations are treated.

Thank you for the suggestion. We have modified the sentence in the revised manuscript as follows:

"*Sarmiento et al. (2004) utilized a tracer named Si* (Si* = Si(OH)$_4$ – NO$_3^-$), which was defined by combining silicate with nitrate…*"

Reviewer#2-8
Line 57: I am not a physicist. By internal tides, do you mean internal tide breaking or isopycnal deformation for the mechanism of mixing?

Yes, it means the breaking of internal tidal waves and isopycnal deformation occurred.

Reviewer#2-9
Line 62: "…North Pacific varies seasonally…"

Thank you for pointing out the mistake. We have corrected it according to the reviewer's suggestion.

Reviewer#2-10
Line 64: "…processes control the seasonal…" What do you mean by variability of the biology? Variability occurs through changing abundance and taxonomy.

Thank you for the comment. We have modified the sentence in the revised manuscript as follows:

"*the seasonal variation of phytoplankton abundance and taxonomy.*"

[Figure]

[Figure]

**Hokkaido University**
Institute of Low Temperature Science

HOKKAIDO
UNIVERSITY

ILTS

Reviewer#2-11
Line 66: What do phytoplankton pigment data indicate? A short explanation could help here.

Thank you for the comment. We have modified the sentence in the revised manuscript as follows:

"*phytoplankton community composition as estimated from the pigment data*".

Reviewer#2-12
Line 135: some typos exist in the supplement (line 41)

Thank you for the comment. In the revised version of the supplement, we have modified the sentence in the revised manuscript as follows:

"*The nutrient contents in the depth range $Z_1$ to $Z_2$ shrink because the input is smaller than the output.*"

Reviewer#2-13
Line 167: instead of "are mainly contributed by" you could write "the main components derive from"

Thank you for the suggestion. We have modified the sentence in the revised manuscript as follows:

"*the upper NPIW (U-NPIW) and the lower NPIW (L-NPIW), whose main components are derived from the Okhotsk Sea water (OSW) and the Western Subarctic Gyre (WSAG) water, respectively.*"

Reviewer#2-14
Figure 2: I think it would be great if you indicated on these hydrographic sections where the main water masses are located with depth.

Thank you for the suggestion. We consider that the salinity minimum feature shown in Figure 2b indicates the presence of NPIW, and that is written in line 171. However, the

depth of NPIW is variable between areas, so we cannot indicate the water masses by depth.

According to the reviewer's comment, we have also added the following sentence to the caption of Figure 2 in the revised manuscript:

*"In Figure 2b, the salinity minimum between the 26.5 - 27.0 $\sigma_\theta$ indicates the presence of U-NPIW. These figures were drawn using the ODV."*

Reviewer#2-15
Line 187: How is the "surface layer" defined?

Thank you for the comment. The "surface layer" was defined by the isopycnal surface 22.0 $\sigma_\theta$, as indicated in Figure 3.

Reviewer#2-16
Line 209: Is this a known behavior for nitrate? Nitrate will act conservatively along a subsurface flow-path only if there are no biological processes occurring, or regeneration is balanced by uptake.

Thank you for the comments. The formation and distribution of nutrient-rich NPIW strongly affect the intermediate water in the North Pacific Ocean (Nishioka et al., 2020). Based on the distribution of NPIW, physical nutrient supply and biological nutrient uptake/regeneration processes are also crucial for determining nutrient concentrations in the study area.

Reviewer#2-17
Line 210: What is a "high concentration"? I would include specific values for these concentrations, or concentration ranges. Similarly, how are the high cores defined as both nutrients have red colors in Figure 4 extending to STG at sigma 27.0. Only station 18 looks exceptionally high.

Thank you for the comments. We have defined the high concentration core as nitrate concentration with > 20 μM; therefore, we have modified the sentence in the revised manuscript as follows:

*"A high macronutrient concentration core (i.e., nitrate concentration with > 20 μM) deeper than 26.5 σ$_\theta$, extended to the STG."*

**Reviewer#2-18**
Line 231: It can not be discerned from the current color scheme that a "nitrate flux is downward" in Figure 5. It looks to be around zero in the STG below sigma 27.0.

Thank you for the comment. As the order of nitrate flux in the STG below 27.0 σ$_\theta$ is $10^{-11}$ – $10^{-8}$, it is hard to show the color in the figure, so it seems like the value is zero.

**Reviewer#2-19**
Figure 7: It would be nice to see a line which indicates the depth of maximum diatom-derived chlorophyll-a on the section plot. This could be drawn between profiles for where diatoms are present.

[Figure]

Thank you for the suggestion. The white line has been added to indicate the depth of SDM and a sentence has been added to the caption of Figure 7 in the revised manuscript:

*"Section profiles of (a) diatom-derived Chl a concentration (mg m⁻³), (b) the subsurface diatom-derived Chl a maximum (SDM) concentration (mg m⁻³) at each station. In Figure 7a, the white line indicates the depth of SDM. Figure 7a was drawn using the ODV."*

Reviewer#2-20
Line 276: "…dFe flux…"

Thank you for the comment. We have modified the sentence in the revised manuscript as follows:

*"between dFe flux, macronutrient fluxes, and phytoplankton abundance…"*

**Reviewer#2-21**
**Line 279: What is meant by "…influence the diatom-derived Chl a…"? You could describe the trend in the plot.**

Thank you for the comment. We have modified the sentence in the revised manuscript as follows:

*"Based on the values of $r^2$ and p-values, dFe and Si fluxes are more highly correlated with diatom-derived Chl a concentration than nitrate and phosphate, indicating that dFe and Si fluxes had higher impacts on the diatom abundance."*

**Reviewer#2-22**
**Line 283: In fact, I think these were diatoms which responded to the Fe additions (silicate was drawn down). A feature that is often seen in HNLC and HNLC-like waters.**

Thank you for the comments. We have cited some previous studies, including Fe fertilization experiments and Fe-addition bottle incubation experiments, which indicated that dFe addition stimulated the diatom growth and supported our conclusions.

**Reviewer#2-23**
**Line 285: "…diatoms form silicified frustules…"**

Thank you for the comment. We have corrected it accordingly.

**Reviewer#2-24**
**Line 314: Please indicate what is meant by "nitrate plus nitrite".**

Thank you for the comment. In this study, we used nitrate concentration solely to calculate the nitrate flux. On the other hand, Endo and Suzuki (2019) used "the combined concentration of nitrate and nitrite" for their discussion.

**Reviewer#2-25**

Line 338: It is well-known that diatoms occupy high nutrient regimes due to their competitive advantage over slow-growing small cells. Diatoms have high affinity uptake of nitrate and have many strategies to store nutrients when available. I wonder if you could include some discussion of how *pro* occupies oligotrophic conditions (where negative flux can occur) due to their ability to deal with lower nutrients through small cell size, slow growth rates, and lower nutrient requirements. Some discussion of these well-known features which drive phytoplankton distributions would provide more context of these great findings. Few studies connect supply ratios with phytoplankton, which is more indicative of the conditions phytoplankton experience (compared to static concentration measurements).

Thank you for the comments and suggestions. We have added the following sentences to the end of this paragraph in the revised manuscript:

*"Additionally, Prochlorococcus are generally expected to deal with lower nutrients due to their small cell size, lower nutrient requirements, and lower maximum growth rates (Patensky et al., 1999). The ecophysiological features of Prochlorococcus could make them more adapted or acclimated in the N-limited STG (Fig. 6a) with lower nutrient conditions (Fig. 4a-d)."*

**References**

FAO: The State of World Fisheries and Aquaculture 2020. Sustainability in action, FAO, Rome, Italy, 244 pp., ISBN 9789251326923, 2020.

Partensky, F., Hess, W. R., Vaulot, D.: *Prochlorococcus*, a marine photosynthetic prokaryote of global significance. Microbiol. Mol. Ecol. Rev., 63, 106-127, https://doi.org/10.1128/mmbr.63.1.106-127.1999, 1999.

End

---

## Referee Report (RR1)

The authors addressed all the comments raised in the previous round of review. I only have a few minor technical comments:

Line 35: I would change the 'with' in the sentence, to make the point clearer. 'Therefore, …. supply and the associate response of…' or something toward this direction

Line 38: '… of both.' Could remove the "aerosol Fe deposition" and the "oceanic Fe supply", already mentioned above.

Paragraph 2 of the introduction would need more attention in terms of writing, to make it more fluid as this part got quite some corrections.

Line 250 – Part 3.5 – maybe could be better combined as it is a bit redundant with the comparison with Kaneko study (line 250 and 253)

Line 294: Fe fertilization instead of iron fertilization, I would add in situ to be clearer Problem with the overall sentence, need to add after '…bioavailable Fe assays

(Nodwell and Price, 2001; Hassler and Schoemann, 2009), were performed and proved the role of Fe for diatom growth. '

This paragraph would need more attention in terms of writing, to make it more fluid as this part got quite some corrections.

Line 394: …' to THE understanding of global oceanic biogeo…' or 'to understand…'

---

## Author Response (AR2)

**Hokkaido University**
Institute of Low Temperature Science

**Response to Editor's comments:**

I am pleased to inform you that both reviewers recommend publication of your revised manuscript. Please consider the technical comments provided by referee 1 when preparing the final version of your article. Thank you for submitting your work to Biogeosciences.

Thank you for considering our revised manuscript for publication in Biogeosciences. We sincerely appreciate the constructive feedback provided by the reviewers and are pleased that both reviewers have recommended our manuscript for publication.

We have carefully addressed the technical comments from Referee 1 and incorporated the necessary revisions into the final version of our manuscript. Please find the updated manuscript and a point-by-point response detailing the modifications.

**Response to the comments from Reviewer #1:**

**General comments**

The authors addressed all the comments raised in the previous round of review. I only have a few minor technical comments.

Thank you for your positive feedback and for recognizing our efforts in addressing the previous comments. We appreciate your careful review and the additional technical suggestions. We have carefully considered and incorporated the necessary revisions in response to your comments. We truly value your time and constructive feedback, which have helped improve our manuscript.

**Specific and Technical comments**

**Reviewer#1-1**

[Figure]

[Figure]

**Hokkaido University**
Institute of Low Temperature Science

Line 35: I would change the 'with' in the sentence, to make the point clearer. 'Therefore,··· supply and the associate response of···' or something toward this direction.

Thank you for the suggestion. We have modified the sentence on line 35 in the revised manuscript as follows:

*"Therefore, it is essential to know the mechanisms of Fe and macronutrient supply and the associate response of phytoplankton communities."*

Reviewer#1-2
Line 38: '··· of both.' Could remove the "aerosol Fe deposition" and the "oceanic Fe supply", already mentioned above. Paragraph 2 of the introduction would need more attention in terms of writing, to make it more fluid as this part got quite some corrections.

Thank you for the suggestion. We have modified the sentence on line 37 in the revised manuscript as follows:

*"We need a coherent explanation for the biological response in the western North Pacific waters that incorporates knowledge of both."*

We have modified sentences on line 35 and line 37 in the paragraph 2 of the introduction according to the comments. We hope the revisions will satisfactorily meet your expectations.

Reviewer#1-3
Line 250 - Part 3.5 - maybe could be better combined as it is a bit redundant with the comparison with Kaneko study (line 250 and 253)

Thank you for the suggestion. We have modified the sentences in the revised manuscript as follows:

In line 249:
*"Consequently, they concluded that there was no clear relationship between the nitrate flux and Chl a concentration."*

[Figure]

[Figure]

**Hokkaido University**
Institute of Low Temperature Science

In line 251:

*"As a result, there is nearly no correlation between total Chl a concentration and individual fluxes or flux ratios (Supplement Fig. S4-S6). However, as the dominant phytoplankton communities would vary between the subarctic and subtropical Pacific, relationships between the phytoplankton community composition and dFe or macronutrient fluxes would attract more attention."*

Reviewer#1-4
Line 294: Fe fertilization instead of iron fertilization, I would add in situ to be clearer. Problem with the overall sentence, need to add after '…bioavailable Fe assays (Nodwell and Price, 2001; Hassler and Schoemann, 2009), were performed and proved the role of Fe for diatom growth.' This paragraph would need more attention in terms of writing, to make it more fluid as this part got quite some corrections.

Thank you for the suggestions. We have modified the sentences on line 293 in the revised manuscript as follows:

*"Since then, a number of ocean in situ Fe fertilization experiments (Coale et al., 1996; Boyd et al., 2000; Tsuda et al., 2003; Boyd et al., 2004; Coale et al., 2004) and Fe-addition bottle incubation experiments (Martin et al., 1990; Hutchins and Bruland, 1998; Nodwell and Price, 2001; Hassler and Schoemann, 2009; Nishioka et al., 2009) were performed. These bioavailable Fe assays proved the role of Fe for diatom growth."*

We have modified sentences on line 294 in this paragraph according to the comments. We hope the revisions will satisfactorily meet your expectations.

Reviewer#1-5
Line 394: … 'to THE understanding of global oceanic biogeo…' or 'to understand…'

Thank you for the comments. We have modified the sentence on line 394 in the revised manuscript as follows:

[Figure]

[Figure]
 Hokkaido University
Institute of Low Temperature Science

*"Our findings contribute to understand oceanic biogeochemical circulation in the North Pacific."*

Other corrections

We have revised the color schemes of Fig. 6 to improve accessibility for readers with color vision deficiencies, ensuring they can correctly interpret our findings.

End

[Figure]

[Figure]

Hokkaido University
Institute of Low Temperature Science

**(Author's tracked changes as below)**

[revised manuscript text omitted]

a **Subsurface chlorophyll *a* biomass maximum of each phytoplankton group**

STG | KE | KOTA | SAG

st.22  21  20  19  18  17  16  15  14  13  12

- Diatoms
- Prasinophytes
- Haptophytes
- Cyanobacteria
- Chrysophytes
- Dinoflagellates
- *Prochlorococcus*
- Chlorophytes

Other plankton concentration (mg m$^{-3}$)

Cholorophytes and *Prochlorococcus* concentration (mg m$^{-3}$)

b **dFe and macronutrient indiv-flux**

dFe flux (µmol m$^{-2}$ s$^{-1}$)
PO$_4$ flux (mmol m$^{-2}$ s$^{-1}$)

NO$_3$ Si(OH)$_4$ flux (mmol m$^{-2}$ s$^{-1}$)

- dFe flux
- PO4 flux
- NO3 flux
- Si(OH)4 flux

c **dFe and macronutrient flux ratio**

N/P and Si/N flux

dFe/Si and dFe/N flux

- N/P
- Si/N
- dFe/N
- dFe/Si

Latitude
10°N    20°N    30°N    40°N

[Figure]

[Figure]

Hokkaido University
Institute of Low Temperature Science

[revised manuscript text omitted]